# Large Extracellular Vesicle Characterization and Association with Circulating Tumor Cells in Metastatic Castrate Resistant Prostate Cancer

**DOI:** 10.3390/cancers13051056

**Published:** 2021-03-02

**Authors:** Anna S. Gerdtsson, Sonia M. Setayesh, Paymaneh D. Malihi, Carmen Ruiz, Anders Carlsson, Rafael Nevarez, Nicholas Matsumoto, Erik Gerdtsson, Amado Zurita, Christopher Logothetis, Paul G. Corn, Ana M. Aparicio, James Hicks, Peter Kuhn

**Affiliations:** 1USC Michelson Center for Convergent Bioscience, University of Southern California, Los Angeles, CA 90089, USA; anna.sandstrom_gerdtsson@immun.lth.se (A.S.G.); msetayes@usc.edu (S.M.S.); Peymaneh87@gmail.com (P.D.M.); ruizvela@usc.edu (C.R.); anders@bionamic.io (A.C.); rnevarez@usc.edu (R.N.); matsumon@usc.edu (N.M.); erik.gerdtsson@gmail.com (E.G.); jameshic@usc.edu (J.H.); 2Department of Genitourinary Medical Oncology, The University of Texas MD Anderson Cancer Center, Houston, TX 77030, USA; azurita@mdanderson.org (A.Z.); clogothe@mdanderson.org (C.L.); PCorn@mdanderson.org (P.G.C.); aaparicio@mdanderson.org (A.M.A.)

**Keywords:** large extracellular vesicles, oncosomes, CTCs, liquid biopsy, metastatic castrate-resistant prostate cancer, aggressive variant prostate cancer

## Abstract

**Simple Summary:**

Non-invasive, liquid biopsies are an attractive means for tumor diagnosis and monitoring. In addition to DNA and cells, tumors have an increased propensity compared to normal cells to shed vesicles. These large extracellular vesicles, which are believed to be more frequent in aggressive cancers, carry tumor DNA and proteins and can thus be informative sources for diagnosis and prognosis. In this study, we developed a method to identify and molecularly characterize large extracellular vesicles in parallel to circulating tumor cells at a single-cell/single-vesicle level. We show that the number of large extracellular vesicles correlates to and exceeds the number of circulating tumor cells, and that analysis of tumor-derived large extracellular vesicles hence increases the sensitivity of the liquid biopsy assay.

**Abstract:**

Liquid biopsies hold potential as minimally invasive sources of tumor biomarkers for diagnosis, prognosis, therapy prediction or disease monitoring. We present an approach for parallel single-object identification of circulating tumor cells (CTCs) and tumor-derived large extracellular vesicles (LEVs) based on automated high-resolution immunofluorescence followed by downstream multiplexed protein profiling. Identification of LEVs >6 µm in size and CTC enumeration was highly correlated, with LEVs being 1.9 times as frequent as CTCs, and additional LEVs were identified in 73% of CTC-negative liquid biopsy samples from metastatic castrate resistant prostate cancer. Imaging mass cytometry (IMC) revealed that 49% of cytokeratin (CK)-positive LEVs and CTCs were EpCAM-negative, while frequently carrying prostate cancer tumor markers including AR, PSA, and PSMA. HSPD1 was shown to be a specific biomarker for tumor derived circulating cells and LEVs. CTCs and LEVs could be discriminated based on size, morphology, DNA load and protein score but not by protein signatures. Protein profiles were overall heterogeneous, and clusters could be identified across object classes. Parallel analysis of CTCs and LEVs confers increased sensitivity for liquid biopsies and expanded specificity with downstream characterization. Combined, it raises the possibility of a more comprehensive assessment of the disease state for precise diagnosis and monitoring.

## 1. Introduction

Tumors frequently shed biological material into the circulation that can be interrogated as surrogate, non-invasive diagnostic or prognostic markers. These include circulating tumor cells (CTCs), vesicles, soluble proteins and cell free DNA. Extracellular vesicles (EVs) are secreted by all cells but at significantly higher rates by cancer cells. Tumor- derived EVs are present in a wide size range and are considered vehicles for crosstalk between both tumor cells and cells of the surrounding microenvironment. The EV payload of proteins, lipids, mRNAs, microRNAs, and DNA is postulated to facilitate regulation of the microenvironment in favor of metastasis [1]. Since they can be targeted and studied in clinically useful minimally invasive liquid biopsies, EVs have proposed implications as diagnostic and prognostic biomarkers, as well as therapeutic targets [2,3,4,5].

Different types of EVs have been described and classified mainly according to size and mechanism of cellular release. While exosomes (30–150 nm) are generated through exocytosis of vesicular bodies, much larger EVs, usually reported in the 0.1–10 µm range and defined by a variety of terminologies including oncosomes [6,7], are generally believed to be assembled at and released from the plasma membrane, carrying a load of proteins and tumor DNA reflecting the tumor they are derived from [8,9].

EVs are most commonly isolated by centrifugation with the disadvantages of low throughput and poor specificity with potential contamination of proteins and non-exosomal particles that may hamper their analysis as diagnostic biomarkers. Recently, promising advances in microfluidics techniques has enabled on-chip capture of EVs through enrichment of specific markers [10]. We present a non-enrichment method for identification of large EVs (LEVs) as a side product of a previously validated single-cell analysis technology enabling molecular characterization of LEVs alongside the CTC in liquid biopsies. The High-Definition Single Cell Analysis (HDSCA) assay has previously been described for CTC characterization for a variety of indications [11,12,13,14]. In the HDSCA protocol, cells are plated on standard format microscope slides, which can be readily integrated with downstream analysis including single-cell genomic and multiplex targeted proteomic analysis [15,16]. Here, we have developed an algorithm to identify LEVs as non-nucleated, membranous CK-positive objects on HDSCA preparations, with subsequent validation through higher resolution imaging and multiplexed protein profiling through imaging mass cytometry (IMC). This approach gives us the ability to revisit any given region of interest (ROI) on a slide that has been stained and scanned from our biobank of >10,000 samples, enabling identification of novel potential biomarkers as well as increased sensitivity of the HDSCA liquid biopsy assay.

Although present in different cancers, LEVs have been frequently identified and described in metastatic prostate cancer [17,18], where it has been shown that they are shed by blebbing of the plasma membrane of amoeboid tumor cells, giving rise to a motile and invasive phenotype [19]. Here, we have quantified and characterized LEVs and CTCs in aggressive variant prostate cancer (AVPC), a subset of metastatic castrate-resistant prostate cancers (mCRPC) of heterogeneous morphologies, that share the atypical and virulent clinical features and therapy response profiles of the small cell or poorly differentiated neuroendocrine prostate carcinomas [20,21]. We show that identification of LEVs may provide increased sensitivity in liquid biopsy assays for AVPC, and that the protein content of single LEVs and CTCs can be analyzed using IMC for delineating combined biomarker profiles of tumor-derived circulating objects.

## 2. Materials and Methods

### 2.1. Clinical Samples

We revisited scanned slide images of samples collected from 44 mCRPC patients (Table 1) who participated in trial NCT01505868. The samples had been processed using the HDSCA protocol as previously described [22,23]. The trial (NCT01505868) evaluated cabazitaxel plus carboplatin relative to cabazitaxel alone [24]. For the purpose of this study, however, only samples collected at baseline were analyzed. In brief, blood samples had been processed within 24 h after collection, with nucleated cells plated in a monolayer ~3 × 10^6^ cells per slide (Marienfeld, Lauda, Germany) before storage at −80 °C. Two slides from each sample had been thawed and stained, using pan-cytokeratin (CK) (1:100; Sigma-Aldrich, St. Louis, MO, USA, Cat#C2562) and CK-19 antibodies (1:100; Dako, Carpinteria, CA, USA, Cat#M0888) with Alexa Fluor 555 secondary antibody (Invitrogen, Carlsbad, CA, USA, Cat#A21127); CD45 (anti-CD45 Alexa Fluor 647–conjugated antibody (1:125; BioRad, Hercules, CA, USA, Cat#MCA87A647X); androgen receptor (AR) antibody ((1:250; Cell Signaling Technology, Danvers, MA, USA, Cat#5153) with Alexa Fluor 488 secondary antibody (Invitrogen, Cat#A11034); and 4′,6-diamidino-2-phenylindole dihydrochloride (DAPI, Cat#D1306, Invitrogen, Waltham, MA, USA), and imaged using automated high-throughput fluorescence scanning.

### 2.2. Identification and Characterization of CTCs and LEVs

Candidate CTCs had been previously quantitated from 10x scan slide images using an algorithm identifying pan-CK-positive and DAPI-positive, CD45-negative events, presented in a report, classified by analyst to different categories, including intact CTCs and apoptotic-like cells, and scored for androgen receptor (AR) positivity. Using the same 3-color immunofluorescence (IF) panel, we developed an image analysis workflow to identify pan-CK-positive, DAPI-negative, CD45-negative round events which were stored in a separate report where they were classified as LEV candidates after visual inspection. Coordinates were stored for all candidate events so that CTCs and LEVs could be reimaged at 40× to confirm or adjust the classification. Brightfield imaging at 40× magnification was used in addition to the IF channels to verify a membranous vesicular structure of the LEVs. Morphometric data on e.g., size and signal intensity were stored for each object, enabling characterization of the different events. Two slides per patient from 44 patients were evaluated and used to calculate correlation of CTCs and LEVs, using linear regression. CTC/LEV correlation to clinical parameters was assessed using Pearson correlation. A total of 399 CTCs, LEVs and apoptotic-like cells identified in 12 positive patients were used to compute size distributions of the different classes of events. 

### 2.3. Imaging Mass Cytometry

A panel of 33 protein markers was designed and used for IMC (Appendix A). Maxpar antibodies validated for IMC use were purchased from Fluidigm (South San Francisco, CA, USA). Antibodies that were not available as Maxpar reagents were conjugated using a kit from Fluidigm, and (for markers applicable) validated using LNCaP and/or PC3 cells spiked in whole blood and processed according to the HDSCA protocol (Appendix A).

Samples from three patients with high concentrations of LEVs, CTCs and apoptotic-like cells were selected for IMC analysis. One fresh HDSCA slide from each patient was thawed from the −80 °C archive, stained with standard 3-color IF protocol for pan-CK, DAPI and CD45 and scanned, classified and reimaged as described above. The slides were subsequently stained with a cocktail of the 33 metal-conjugated antibodies using optimized dilutions for each reagent, and a staining protocol previously optimized for HDSCA slides [16].

CTCs and LEVs confirmed by 40× imaging were relocated in the Hyperion IMC instrument (Fluidigm, South San Francisco, CA, USA) using a defined offset between coordinates of the 80i microscope (Nikon, Melville, NY, USA) and the Hyperion stage. Each cell/LEV of interest was centered in a region of interest (ROI) of 200 × 200 µm. Using automated batch mode, the ROIs were laser ablated at 200 Hz, in a raster pattern with 1 µm^2^/pulse being aerosolized followed by ionization and mass quantitation in the CyTOF Helios unit. For each ROI, one image per antibody ‘channel’ was rendered where the intensity of each pixel corresponded to the ion count of the respective laser ablation pulse. An in-house software application was used to analyze the images, which were also incorporated in the corresponding original slide report for comparison to the IF 40× image of each cell/LEV of interest. Unsupervised hierarchical clustering and tSNE analysis was used to assess relatedness of biomarkers and object categories.

## 3. Results

### 3.1. Identification and Morphometric Characterization of LEVs

We developed a method for quantification and characterization of LEVs, integrated with the HDSCA workflow (Figure 1A). To identify LEVs, an algorithm was trained to find non-nucleated vesicular objects defined as DAPI-negative, pan-CK-positive, round events, on 10× scan images that had been processed using the HDSCA workflow. A report system was created, enabling preliminary classification of objects as candidate LEVs or ‘junk’. For each slide, coordinates of candidate LEVs were added to previously identified nucleated cells of interest. High-resolution (40×) imaging was then used to manually confirm/adjust the computed classification of pan-CK-positive, CD45-negative events, including CTCs (intact cells larger than surrounding WBCs), and apoptotic-like cells (disrupted nucleus and/or blebbing cytoskeleton). Non-nucleated objects with a clear membranous structure as verified with brightfield imaging were classified as LEVs (Figure 1B).

In CTC and LEV-positive samples, the average diameter of the LEVs was 9.8 (±3.0) µm, ranging from 6.3 to 20.4 µm (Figure 1C). Smaller pan-CK-positive, DAPI-negative objects (<6 µm) were often present in high numbers but omitted as it was deemed that membranous structures could not be confirmed in the report based on 10× magnification scans. CTCs appeared in a wide size distribution (15–45 µm) with an average diameter of 25.3 (±5.9) µm. The size distribution of apoptotic-like cells was in between that of LEVs and CTCs, with an average diameter of 13.5 (±3.7) µm, ranging from 6.7 to 25.1 µm.

Based on two slides, corresponding to approximately 6 × 10^6^ nucleated cells interrogated per sample, CTCs were identified in 21/44 (48%) of patients, and LEVs in 35/44 (80%). Seven patients (16%) were negative for both CTCs and LEVs. Average coefficient of variation between two slides from the same patient were 0.39 and 0.21 for CTCs and LEVs, respectively, calculated for patients with >5 events in average. Albeit skewed towards low prevalence, linear regression (R = 0.95, *p* < 2.2 × 10^−16^) also showed that LEVs were present at higher frequencies than CTCs, with 1.9 as many LEVs than CTCs being identified (Figure 1D). Within CTC-negative patients (23/44, 52%), at least one LEV was identified in 73% (17/23) of cases demonstrating that adding LEV detection in parallel to the CTC classifier resulted in improved sensitivity of the liquid biopsy. Neither CTC, LEV, nor combined events were significantly associated to any of the clinical parameters tested (Appendix A), however a tendency for higher levels of all object classes were seen for higher tumor load and bone marrow positive cases (Figure 1E). In summary, strict inclusion/exclusion criteria based on fluorescence markers followed by visual inspection of high-resolution images, correlation to CTC, and strong sample-dependent presence, indicate that the LEVs are tumor-associated events.

### 3.2. Protein Profiling of CTCs and LEVs

A biomarker panel of 33 antibodies for protein profiling was designed to evaluate prostate cancer invasiveness, epithelial-to-mesenchymal transitioning (EMT), cell cycle proliferation and apoptosis, stem cell-ness, and discrimination of EV classes (Appendix A). To evaluate the panel for IMC analysis, samples from three patients with high concentrations of CTCs, LEVs and apoptotic-like cells were analyzed, with a total of 170 ROIs, containing objects of the CTC (*n* = 51), LEV (*n* = 79), and apoptotic (*n* = 40) classes, as well as the surrounding WBCs captured in each 200 × 200 µm ROI.

Overall, the IMC ion count-rendered images correlated well with the corresponding IF images of the same ROIs (Figure 2A) consistent with our previously reported results [16,25]. The CK signal-to-noise was lower in IMC than in IF, which was expected since more CKs were targeted with the IF pan-CK cocktail than the IMC CK8/18 antibody and since the IMC antibody was partly prevented by binding to the IF antibodies masked epitopes. In a subset of LEVs, the DNA intercalator added to the IMC staining protocol revealed a weak signal from nucleic acids where DAPI signal had been absent in the IF analysis, demonstrating that these objects indeed contained small amounts of nucleic acid material (Figure 2B).

IMC signals for each marker and each object were scored on a 0–3 scale based on ion count signal-to-noise for the cell of interest in relation to surrounding WBCs. When excluding DNA signal from the data set, unsupervised clustering of the score data showed that the different classes of objects interrogated were poorly discriminated based on protein profiles (Figure 2C and Appendix A), although differing in the overall protein load, with average sum of scores of the proteins interrogated being 11.9 for CTCs, 8.1 for LEVs and 4.8 for apoptotic-like cells. CTCs and LEVs could thus mainly be discriminated based on size, DNA load, and morphology (Appendix A). Unsupervised clustering by IMC scores was weakly associated to patient, although the separation was not significant based on the proteins assessed (Appendix A).

Albeit the number of patients were too few to draw any conclusions, the IMC analysis revealed markers of interest and indicated subgroups of LEVs and cells. Antibodies against HSPD1, HSPA5 and ATP5B were added to the panel as they had been previously identified specifically in LEVs compared to exosomes, in addition to being associated with prostate cancer progression [26]. HSPD1 showed nuclear presence in 98% of CTCs, 97% of LEVs, and 86% of apoptotic-like cells, with predominantly strong staining (score 2–3), while being absent or low in WBCs (Figure 3A), demonstrating that HSPD1 is a promising marker for specifically identifying and enriching for tumor derived cells and LEVs in prostate cancer liquid biopsies. 

### 3.3. Epithelial and Prostate Marker Distribution 

As many CTC technologies rely on detection/enrichment of either EpCAM−or CK-positive cells, we used the IMC analysis to investigate the expression of both epithelial markers. EpCAM, which clustered with PSA, showed a heterogeneous expression pattern across the objects interrogated. (Figure 3B). Pan-CK was the marker used to select candidate events in the initial IF analysis, thus all objects interrogated by IMC were pan-CK-positive. Among all pan-CK-positive objects (CTCs, apoptotic-like cells and LEVs), 49% were negative for EpCAM, 32% were weakly positive (1+) and only 19% had a 2+ or 3+ score. In addition to pan-CK, the majority of EpCAM-negative objects carried other tumor markers indicating a tumor origin. Among EpCAM-negative, pan-CK-positive events, 38% of CTCs and 46% of LEVs lacked presence of any of the markers PSA, PSMA or AR, but were with the exception of 1 CTC and 2 LEVs, positive for HSPD1. 

Overall, the expression of prostate cancer markers AR, PSA, and PSMA was heterogeneous, as demonstrated by the unsupervised clustering (Figure 2). Among CTCs, 86% were positive for at least one of the markers, while the positivity rate was lower for LEVs (63%) and apoptotic-like cells (45%). Expression of PSMA appeared highly patient dependent with high-scoring CTCs/LEVs observed almost exclusively in Patient 2. 

CTC expression of AR splice variant AR-V7 is a demonstrated predictive marker for mCRPC [27,28,29]. In two of the three patients analyzed by IMC, a total of only five CTCs and LEVs showed a (weak) expression of AR-V7, in all cases co-expressed with AR.

### 3.4. Protein Profile Heterogeneity

The IMC analysis added significant levels of detail to the single cell characterization and revealed a striking heterogeneity among the objects of interest that could not be observed by the high resolution IF images alone. In particular, expression of EMT markers Vimentin and/or Twist-1 was observed in a small cluster of objects (Figure 4A) identified through unsupervised clustering, including events from all three patients. The majority of events displayed concurrent signals from both epithelial markers, indicating a partial EMT profile. Figure 4B shows an example of a CTC expressing epithelial markers CK8/18, EpCAM and E-cadherin in addition to EMT markers Vimentin, Twist-1, as well as several prostate cancer markers, including AR-V7. It also expressed phospo-p38, which has been previously linked to survival and resistance and which was associated with Twist-1 by clustering (Figure 2C).

Additional observations from single-cell protein profiling included markers of stemness, proliferation and apoptosis. Albeit interesting, the total number of objects positive for these markers were few. The cancer stem cell marker CD44 scored positive in a small number (*n* = 8) of (CD24-negative) objects (Figure 2C). Interestingly, the expression of CD44 was exclusive for smaller objects, accordant with previous reports correlating small cell size with stemness [30].

Caveolin-1 (Cav-1), the main component of the cavaeolae plasma membrane, promotes cell cycle progression through the Ras-ERK pathway and was previously identified in cell line models as a marker in aggressive prostate cancer LEVs [31] for discriminating LEVs from exosomes [26]. Specificity of the in-house conjugated Caveolin-1 antibody was confirmed using plated cell lines (Appendix A) but Caveoliin-1 was a poor marker of patient LEVs in our experiment. Eighteen objects (CTCs and LEVs) scored weakly positive for Caveolin-1, and only one LEV was strongly positive. CD59 is a prostasome marker for smaller (40–500 nm) prostate derived EVs, included to investigate its presence in larger vesicles. Seven LEVs showed a weak (1+) CD59-expression, while it was absent in CTCs. Tetraspanins, including CD9, CD63 and CD81, are traditionally used as biomarkers for exosomes. Due to lack of antibodies validated for IMC and failure of attempts to custom conjugate others, only CD9 was included in the present panel. CD9 showed weak expression in a handful of cells, and only in one LEV, further supporting the discrimination of LEVs and exosomes.

Apoptosis marker cleaved Caspase-3 was more frequently expressed in LEVs (19%) than nucleated cells (8%) (Figure 2C). Interestingly, the cells that were positive for cleaved Caspase-3 were all smaller in size than the average CTC, which could be explained by cell shrinkage as an indication of initiation of apoptosis. The majority of cells that had been classified as apoptotic-like due to blebbing/retracted cytoskeleton were however negative for cleaved Caspase-3.

## 4. Discussion

The frequent low abundance of CTCs and, for certain tumors, low rate of CTC-positive patients, have hampered the general utility of CTC assays in the clinical setting. By simultaneous identification and profiling of CTCs and LEVs we showed that LEVs were present at close to twice the frequency of CTCs, and in 73% of CTC-negative patients, conferring a significantly higher sensitivity to the liquid biopsy assay in the AVPC cohort. This becomes particularly important in the context of the third-generation comprehensive liquid biopsy in which we combine cell-free DNA analysis with LEV analysis and CTC analysis using both targeted proteomic and genomic approaches. Thus, with the now established HDSCA workflow, three informative sources of tumor-derived material can be readily analyzed from one tube of blood.

It has been previously suggested that the molecular content of tumor-derived vesicles can be very heterogeneous [32], highlighting the benefit of using single-object analysis for their characterization. This also enables more refined biological assessments, e.g., exploring subpopulations in the circulation, and moreover limits the contamination of vesicles shed by non-cancer cells observed in bulk analysis [33]. In comparison to previous studies where CTC technologies have been employed to study circulating vesicles [34], our approach offers higher resolution imagery and high sensitivity as well as downstream multiplexed protein profiling capability. Microfluidic technologies have also been developed for single vesicle analysis, but with less multiplexing capability and lack of parallel analysis of CTCs [35,36].

The standard slide format used within the HDSCA has enabled integration of two modules for downstream single-cell characterization: whole genome copy number variation profiling [15] and imaging mass cytometry (IMC)-based protein profiling [16]. In the present study, we performed IMC analysis on three patient samples with high numbers of LEVs, CTCs and apoptotic cells. To this end, a panel of 33 antibodies was designed, to our knowledge, the most multiplexed protein profiling of vesicles at the single-object level performed to date. With the exception of the DNA stain, CTCs, LEVs, and apoptotic-like cells were not discriminated based on the protein panel used. The panel was carefully selected to reflect not only vesicular biomarkers and oncoproteins previously shown to be carried by LEVs [26,37], but also tumor related biological processes and prostate specific biomarkers. Several markers showed expression in only a small subset of objects, which was interesting per se as rare processes such as (partial) EMT and heterogeneity in prostate cancer specific markers could be captured. Capturing weakly expressed markers with single-object analysis is challenging, and several of the interrogated proteins might have expression levels below the limit of detection of IMC. It remains to be assessed whether the presumably low genomic content (based on the weak to non-existing DNA stain) of the LEV will suffice for genomic profiling. However, as the IMC-analysis disrupt the cells, genomic analysis will have to be performed on sister slides and will thus not allow for correlation between genomic and protein markers.

Many CTC assays, including CellSearch, rely on EpCAM enrichment for identification of tumor derived cells. Our study showed that 49% of the pan-CK-positive objects were EpCAM-negative, of which the majority expressed other tumor markers in addition to pan-CK, including AR, PSA and PSMA. In this context, HSPD1 (HSP60) also emerged as a promising marker, specifically expressed at high levels in CTCs and LEVs, while absent or low in leukocytes. HSPD1 is a mitochondrial chaperone involved in anti-apoptosis and prostate cancer progression. It has been previously shown that LEVs are rich in HSPD1 as well as several other heat-shock proteins (HSPs), commonly expressed in response to cellular stress [26,38]. In pathological conditions, HSPs can be extracellularly transported through EVs and thus be present in the cell membrane [39]. Consequently, HSPs have successfully been used for isolation of EVs [40]. In the present study, the location of HSPD1 appeared exclusively intracellular/nuclear in the CTCs as revealed by the high-resolution IMC rendered images. Other HSPs may be included in future biomarker panels to investigate their presence, location and potential role as biomarkers or tentative targets in CTCs and LEVs from different cancer types. Further IMC analysis of a larger number of patients will reveal whether subsets of events based on biomarker profiles can be correlated to clinical features.

The LEVs identified here had an average size of 9.8 µm, ranging from 6.3 µm to 20.4 µm, which is larger than what has been reported in general for LEVs or large oncosomes. Smaller (<6 µm) pan-CK-positive, DAPI-negative objects were frequently observed but not included in the analysis as membranous structures could not be confirmed at 10× resolution. This will motivate the use of higher resolution imagery in future generation assays, potentially with fully automated classification performed directly in the scanner, which will omit the need for the additional re-imaging step and increase the size range and number of identified vesicles.

A continuum of cell morphologies, from intact CTCs displaying a broad size distribution, to smaller apoptotic-like cells, was observed in AVPC. The apoptotic-like cells were classified on basis of speckled pan-CK staining and/or blebbing nucleus, features that are likely the result of cytoskeletal retraction and degradation associated with the apoptotic cascade. While it has been demonstrated that the formation of large oncosomes occurs through non-apoptotic blebbing [19], it has also been argued that some of the LEVs present in aggressive cancers may be apoptotic vesicles secreted by (apoptotic) CTCs or cells of the solid tumor by membrane bubbling occurring during the apoptotic process [41]. Both LEVs and apoptotic bodies have been described, with the approximate same size range. The possibility also exists that the observed objects may be tumor cells that have lost their nuclear envelope through mechanisms likely related to apoptosis, including retraction of cytoskeletal proteins. In addition, the presence of so-called micronuclei has been reported as small bodies with fragmented genomic material frequently observed in cancers with high chromosome instability [42]. The IMC analysis indeed demonstrated that several of the LEVs showed a (weak) signal from the DNA intercalator, indicating that genomic material was contained by at least a subset of the identified objects. Despite having imaged thousands of cells and LEVs to date, we have yet no examples of objects caught in the processes of either nuclear extrusion or membrane blebbing. The low expression of cleaved Caspase-3, and poorly discriminated protein profiles of LEVs and CTCs observed in our study suggest that the vesicles observed in the present AVPC cohort were derived from solid or circulating viable tumor cells.

The propensity of vesiculation varies significantly between different tumor types and molecular subtypes of cancer [32]. In line with previous reports, our unpublished observations from other cancer types suggests that the frequency of LEVs is a particularly strong feature of aggressive tumors. Based on the restricted number of patients included in this study, data indicated that levels of LEVs and CTCs are associated with tumor load and bone marrow involvement in AVPC. Although larger cohorts will be needed to assess correlation to these and other clinical factors, the results imply that highly sensitive liquid biopsy assays can be developed for both differentiating tumor subtypes and monitoring disease of particularly aggressive cases. AVPC, defined by clinicopathological [20] and/or molecular criteria [21], needs to be identified as patients may benefit from intensified treatment e.g., addition of carboplatin to cabazitaxel [24]. A molecular profile composed of combined immunohistochemistry and/or genomic defects in the tumor suppressors TP53, RB1 and PTEN has been described [21] but the methodology to optimally identify this profile remains to be defined and further molecular characterization is warranted to account for the heterogeneity within the group. The power of EV analysis has already been demonstrated, e.g., for monitoring of glioblastoma therapy [43]. The ability of parallel, single-object CTC and LEV analysis described in our study will further increase the potential of such applications, particularly combined with cell-free DNA analysis. Such comprehensive liquid biopsy analysis could be valuable as minimally invasive means for diagnosis of e.g., AVPC.

## 5. Conclusions

Taken together, the present study demonstrates that identification and characterization of LEVs alongside CTCs at a single-object level confers increased sensitivity and additional biomarkers of the liquid biopsy, which we believe carries vast potential for future clinical implications.

## Figures and Tables

**Figure 1 cancers-13-01056-f001:**
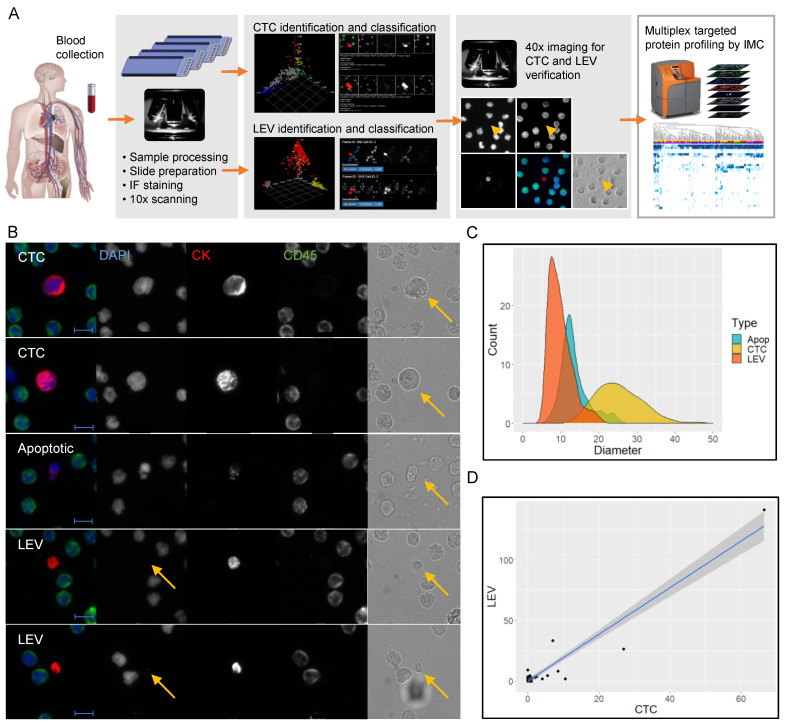
Identification and size determination of circulating tumor cells and LEVs. (**A**) The integrated LEV/HDSCA workflow. (**B**) Examples of objects identified in one patient, classified to different categories and verified with higher resolution (40×) immunofluorescence and brightfield imaging. Composite shows DAPI in blue, pan-CK in red and CD45 in green. Scale bar is 10 µm. Arrows points to cell or LEV (DAPI-negative) for clarification. (**C**) Size distribution of LEVs, CTCs and apoptotic-like cells. (**D**) Linear regression of LEVs and CTCs with number of objects identified per slide, averaged over 2 slides analyzed per sample. (**E**) Cell/LEV enumeration grouped by tumor load (left panel) and bone marrow status (right panel). Open circles represent individual datapoints. Filled black circles and black line represent mean values and standard deviations, respectively.

**Figure 2 cancers-13-01056-f002:**
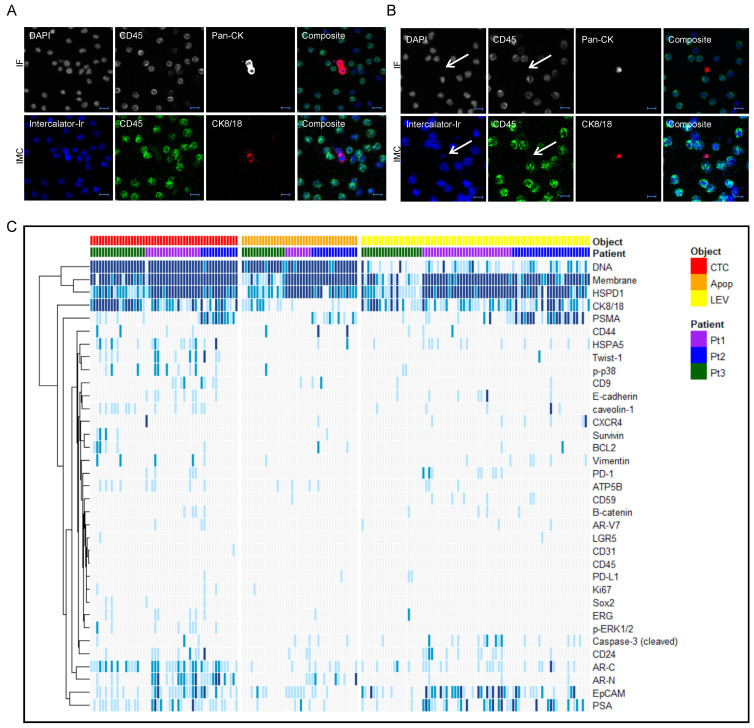
Multiplex protein profiling of CTCs and LEVs. (**A**,**B**) Example of IF (top panels) and IMC (bottom panels) images of a CTC (**A**) and a LEV (**B**). The arrow points at the LEV which is negative for DAPI and weakly stained by the IMC DNA intercalator. Scale bar is 10 µm. (**C**) Heatmap of IMC scores on a 0–3 scale with 0 = white and dark blue = 3, ordered by object categories and patients, and clustered by proteins.

**Figure 3 cancers-13-01056-f003:**
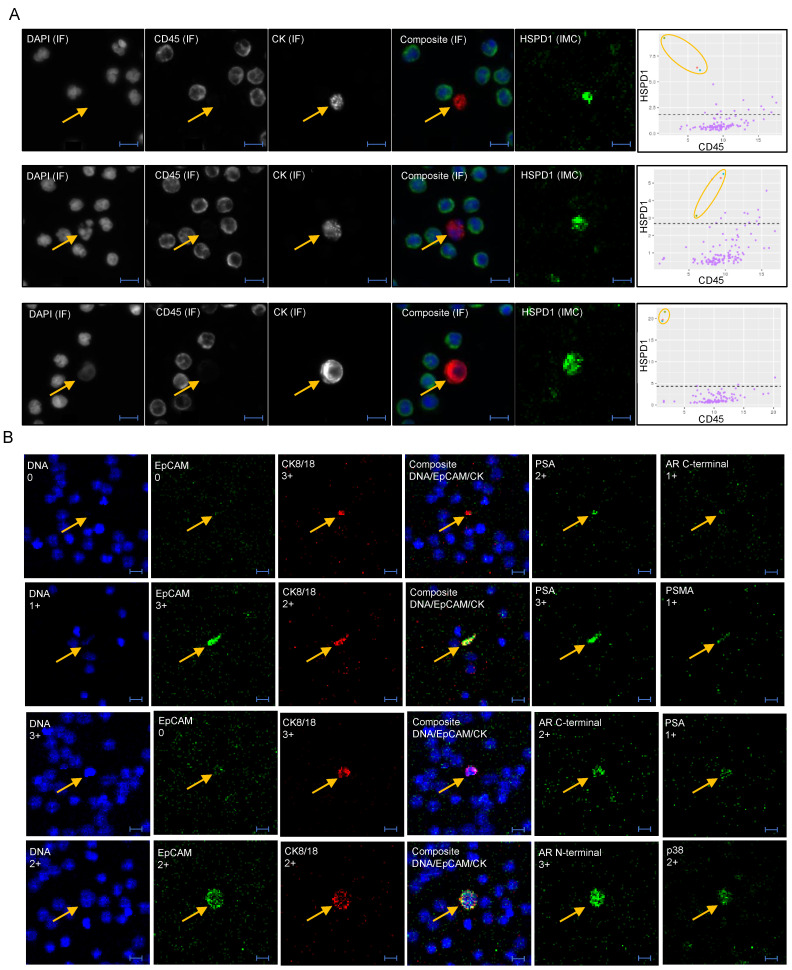
Biomarker expression in CTCs and LEVs of AVPC. (**A**) Example of HSPD1 expression in LEV (top panel), apoptotic (middle panel) and CTC (bottom panel). Cropped ROI images are shown with individual IF channels and IF composite image with DAPI (blue), CD45 (green) and pan-CK (red). The IMC rendered image of HSPD1 signal is shown in green. Table 1 vs. CD45 for all objects within the full ROI with WBCs in purple and the cell of interest (circled) in red (full cell), green (cytoplasm) and teal (nucleus). (**B**) Selected biomarkers shown for LEVs (top 2 panels) and CTCs (bottom 2 panels), with EpCAM expression varying from negative (0) to strong (3+). Scores from the 0–3 scale is shown for each marker and object. Scale bar is 10 µm.

**Figure 4 cancers-13-01056-f004:**
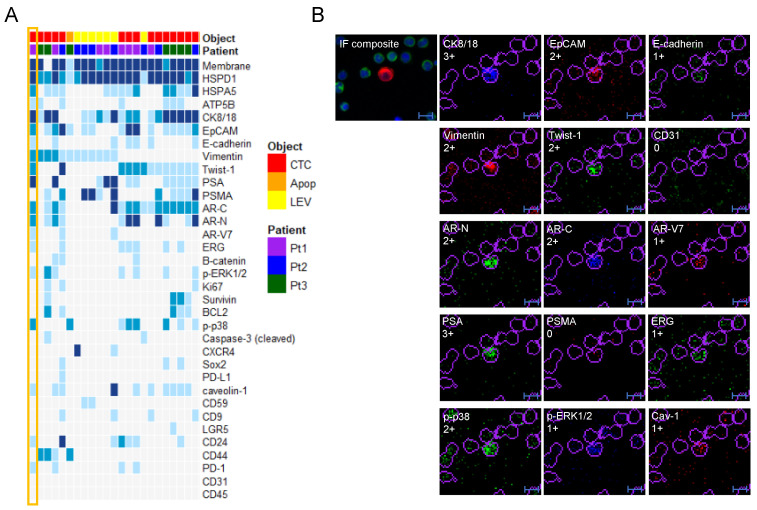
Concurrent expression of EMT and epithelial markers detected by IMC analysis. (**A**) Subset of objects positive for EMT markers Twist-1 and Vimentin, most with concurrent expression of epithelial markers (EpCAM, CK8/18, E-cadherin). Selected individual channels from the CTC marked in (**A**) is shown in (**B**). In addition to the channels shown, this particular cell scored positive for HSPD1 (3+), HSPA5 (2+), ATP5B (1+), PD1 (1+) and CD24 (1+). CD31 (negative) was included to support that Vimentin-expression cannot be explained by endothelial origin. Scale bar is 10 µm.

**Table 1 cancers-13-01056-t001:** Clinical characteristics.

Feature		Distribution
Age		Median 67.5, Range 56–81
Race	Caucasian	*n* = 36 (82%)
	African American	*n* = 3 (7%)
	Hispanic	*n* = 2 (5%)
	NA	*n* = 3 (7%)
ECOG	0	*n* = 12 (27%)
	1–2	*n* = 29 (66%)
	NA	*n* = 3 (7%)
Clinicopathological AVPC (AVPC-C)	Yes	*n* = 16 (36%)
	No	*n* = 25 (57%)
	NA	*n* = 3 (7%)
Prior therapy	Docetaxel	*n* = 11 (25%)
	Abi. and/or Enza. *	*n* = 28 (64%)
	Other	*n* = 10 (23%)
	NA	*n* = 3 (7%)
Bone marrow positive	Yes	*n* = 15 (34%)
	No	*n* = 22 (50%)
	NA	*n* = 7 (16%)
Tumor load	High	*n* = 23 (52%)
	Intermediate	*n* = 12 (27%)
	Low	*n* = 6 (14%)
	NA	*n* = 3 (7%)
PSA (ng/mL)		median 40.3, range 0.1–681.6
PFS (months)		median 6.1, range 1.1–11.7
OS (months)		median 22.6, range 4.7–39.7

* Abi. = Abiraterone, Enza. = Enzalutamide.

## Data Availability

The data presented in this study are available on request from the corresponding author.

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
