# Peer review of "Large Extracellular Vesicle Characterization and Association with Circulating Tumor Cells in Metastatic Castrate Resistant Prostate Cancer"

_cancers, 2021, doi:10.3390/cancers13051056_

Round 1

Reviewer 1 Report

The manuscript by Gerdtsson et al. describes a novel approach allowing for parallel single-object analysis of CTCs and LEVs in a non-enrichment based liquid biopsy platform to interrogate primary CRPC tumor material. The topic is highly relevant and the manuscript is very well-written. The Results section includes the meticulous description of the integrated platform and workflow used for identification, quantification and morphometric characterization of LEVs. The Authors report a correlation between the number of LEVs and CTCs and propose an increased detection sensitivity with LEVs identified in 73% of CTC-negative samples. The Authors then present a multiplex protein signature analysis by high-resolution IF and subsequent 33-mer IMC in a total of 170 ROI’s from 3 patients with high concentrations of CTC, LEVs and apoptotic cells. The Authors report, while CTCs and LEVs differ in size, morphology, DNA load and overall protein score, they are poorly discriminated by protein signatures. The Authors then interrogate epithelial and prostate biomarker signatures of panCK-positive events. They report, 49% are EpCAM-negative and among those, 38% of CTCs and 46% of LEVs lack prostate signatures but are largely HSPD1 positive. AR-V7 expression was detected in 2 out of 3 patients in a total of 5 CTCs and LEVs.

Overall, the integrated approach to single-object high-resolution characterization and multiplex biomarker profiling presented in this study may provide a strong platform for comprehensive analysis of scarce primary tumor material with potential clinical-translational utility. However, the biomarker profiling only includes 3 patients from an exceptionally diverse disease group, which does not allow for broad conclusions from current data set.

Specific Comments:

  • The clinical-translational utility of CTCs as prognostic and predictive biomarkers has been previously demonstrated by various liquid biopsy platforms in CRPC. In current report, neither CTC, LEV, nor combined events showed association with statistical power with any of the clinical parameters tested. How do the Authors explain the lack of statistical correlations in the current study cohort?

  • The Authors report considerable heterogeneity of EpCAM and AR-signature protein expression within the CTCs and LEVs detected. Have the Authors performed correlation analysis between clinical parameters and any distinct LEV subsets like EpCAM+, EpCAM- or others in any CRPC patient cohort?

  • Have the Authors assessed panCK- CTCs or LEVs in CRPC specimen?

  • In the 3 model patients represented in the protein signature analysis, a considerable subset of both CTCs and LEVs detected were also EpCAM- and AR-signature negative. Have the Authors done any further investigation into the prostate origin of these particular events; including gene expression analysis or genomic biomarker screening? Could some of these events include non-specific objects that were missed by the CD45- exclusion criteria? Specifically, HSPD1 expression patterns represented in Figure 3A, show a considerable variation in CD45 expression levels. While the representative CTC expresses virtually zero CD45, the spectral distribution of CD45 signal for both LEV and apoptotic cell overlaps with CD45low WBCs. How have the Authors set the baseline for CD45 positivity? As in reference to what is described as (manually) “confirm/adjust classification” under Results 3.1 on Page 4? Have the Authors included any technical or biological controls to establish these baselines?

  • Previous studies have suggested, expanding exclusion criteria beyond CD45 expression increases accuracy of CTC detection. While some myeloid WBCs may bind panCK antibody aspecifically, some neutrophils and CD45- immune subsets including a subset of mature plasma cells may also express panCK specifically. This would be particularly important when analyzing/confirming EpCAM- and AR- subsets of LEVs and CTCs. Have the Authors assessed exclusion of any myeloid, endothelial, plasma cells or nucleated ertythroid progenitor markers for LEV detection?

  • How reliable is the detection of 1 LEV in this analysis? How many patients in the cohort represented similarly low yield (<3) of LEV and were also negative for CTC? Was there a sufficient number of total events analyzed to ensure the statistical significance of the data in this rare-event analysis platform with such low expected abundance? What was the distribution of various CTC and LEV counts within the patient cohort? Can the Authors clarify if the data points in Figure 1 show average count between the 2 slides processed for IF or the total of 2 slides? Can the Authors comment on intra-patient reproducibility between the 2 slides processed under Results 3.1?

  • In Figure 1E, do the data points in the Tumor Load and in the BM status graphs represent the same cohort? Do the open circles represent individual data points? There seem to be fewer patients plotted on the BM graphs vs the Tumor load graphs. For example, some of the patients within ~20-30 count range seem to not be represented on the BM status graph. Can the Authors clarify? This figure would benefit from the addition of mean/median + error bars to better reflect distribution.

  • The counts were normalized to approximate number of nucleated cells put on slides and ROIs interrogated. Have the Authors looked at counts normalized per ml blood as the nucleated yields per ml peripheral blood could vary considerably in this patient group?

  • The Authors should make the ‘CK’ abbreviation concise throughout the manuscript. CK is first mentioned in the abstract as reference to ‘cytokeratin’ in context of the IMC platform, then again used as an abbreviation on Page 3 in Materials and Methods 2.1 as reference to ‘pan-cytokeratin’.

  • On Page 3, under Methods 2.2 in the description of CTC and LEV identification, the CTCs were described as ‘pan-CK-positive’ and the LEVs were defined as ‘CK-positive’ events. The Authors should clarify if the criteria were different for the identification of CTCs and LEVs. Figure 1B and legend should also include the key. What were the specific pan-CK and CK19 clones used in the protocol included on page 3?

  • On Page 9, under Results 3.3 the Authors report that ‘49% were negative for EpCAM’. The Authors should clarify what population this data refers to.

  • The IMC analysis presented under Results 3.4 on Page 10 shows a subset of panCK positive events from Patients 1,2,3. The Authors should clarify how this subset was selected for this analysis and how they reflect to the overall heterogeneity in Patients 1, 2 and 3. For example, the Authors report 49% of the events presented in section 3.3 do not express EpCAM, however, only 7 out of 24 objects depicted on Figure 4A seem to be EpCAM negative.

  • There is a typo on Page 5, in second paragraph ‘6x106’.

Author Response

Response to Reviewer 1 Comments

The authors appreciate the reviewer’s positive comments, the meticulous review of the manuscript and many insightful and helpful comments. Regarding the few number of patients included in the IMC analysis, the following was added to the results section (p8): “Albeit the number of patients were too few to draw any conclusions, the IMC analysis revealed markers of interest and indicated subgroups of LEVs and cells.. “

Point 1: The clinical-translational utility of CTCs as prognostic and predictive biomarkers has been previously demonstrated by various liquid biopsy platforms in CRPC. In current report, neither CTC, LEV, nor combined events showed association with statistical power with any of the clinical parameters tested. How do the Authors explain the lack of statistical correlations in the current study cohort?

Response 1:  The authors appreciate the feedback on this point. Previous prognostic and predictive data was in the context of a particular clinical utility. This is a concept study with a limited number of patients and with only 21 and 35 CTC- and LEV-positive cases, respectively. Larger patient cohorts will be needed to investigate the correlation to clinical parameters.

Clarification has been added to Discussion, p13: “Further IMC analysis of a larger number of patients will reveal whether subsets of events based on biomarker profiles can be correlated to clinical features.”

Point 2: The Authors report considerable heterogeneity of EpCAM and AR-signature protein expression within the CTCs and LEVs detected. Have the Authors performed correlation analysis between clinical parameters and any distinct LEV subsets like EpCAM+, EpCAM- or others in any CRPC patient cohort?

Response 2: The protein profiling was in this first study only performed on three patients, thus no correlation of subsets based on biomarker to clinical parameters were performed as those would not be statistically sound.

Point 3: Have the Authors assessed panCK- CTCs or LEVs in CRPC specimen?

Response 3: The initial identification of CTCs/LEVs relies on the HD-SCA assay which is a panCK immunofluorescent staining assay and as a result panCK- cells/vesicles could not be assessed, which is detailed in the manuscript (p8). While the assay has considerable dynamic range, an assay that would detect panCK- events would require a different fluorescent assay based on other reliably measured markers, which is beyond the scope of the current study.

Point 4: In the 3 model patients represented in the protein signature analysis, a considerable subset of both CTCs and LEVs detected were also EpCAM- and AR-signature negative. Have the Authors done any further investigation into the prostate origin of these particular events; including gene expression analysis or genomic biomarker screening? 

Response 4: There is substantial heterogeneity across protein expression patterns across most proteins and this observation was not unexpected. As the HD-SCA assay is a single cell methodology using fixed cells, gene expression analysis would be challenging and has not been developed as a downstream analysis option. Copy number variation profiling is regularly performed on CTCs identified through HD-SCA but the missing (or very low level) signal from DNA staining, single LEVs did not generate reliable genomic data. In addition, the IMC profiling disrupt the cells, hence genomic analysis would have to be performed on sister slides, without parallel information on e.g. EpCAM expression.

This has been added to discussion (p12): “It remains to be assessed whether the presumably low genomic content (based on the weak to non-existing DNA stain) of the LEV will suffice for genomic profiling. However, as the IMC-analysis disrupt the cells, genomic analysis will have to be performed on sister slides and will thus not allow for correlation between genomic and protein markers.”     

Point 5: Could some of these events include non-specific objects that were missed by the CD45- exclusion criteria?

Response 5: The HD-SCA algorithm is based on machine learning using a combination of measurements from pan-CK, DAPI, and CD45-signal as well as morphology. Each event is also visually inspected in a 4-channel report system to ensure CD45-negativity and exclusion of false positives. For CTCs, the approach has been validated using e.g. genomic profiling of candidate cells. While genomic profiling has not been performed for LEVs due to the lack of sufficient genetic material (DAPI-), with the strict criteria for their inclusion/exclusion, i.e. strong pan-CK-staining and negative CD45 together with confirmation using high-resolution imaging which clearly discriminated LEVs from leukocytes, we deem that it is highly unlikely that they would be CD45 (false) positives. In addition, their correlation to CTCs and their highly sample dependent presence also point towards a valid algorithmic approach.

We have clarified how they are classified (including visual inspection) in the methods section (p3) and added language in result section (p5): “In summary, strict inclusion/exclusion criteria based on fluorescence markers followed by visual inspection of high-resolution images, correlation to CTC, and strong sample-dependent presence, indicate that the LEVs are tumor-associated events.”

Point 6: Specifically, HSPD1 expression patterns represented in Figure 3A, show a considerable variation in CD45 expression levels. While the representative CTC expresses virtually zero CD45, the spectral distribution of CD45 signal for both LEV and apoptotic cell overlaps with CD45low WBCs. How have the Authors set the baseline for CD45 positivity? As in reference to what is described as (manually) “confirm/adjust classification” under Results 3.1 on Page 4? Have the Authors included any technical or biological controls to establish these baselines? 

Response 6: Inclusion of LEV candidates is based on the HD-SCA IF assay, where CD45 negativity (defined based on standard deviation over the mean (SDOM) calculations in relation to surrounding leukocytes) is one of the criteria. As all the events interrogated by IMC were also imaged and inspected at high (40X) resolution, we are confident that no CD45 (false) positive events were included in the present analysis. The ion count plot is based on CD45 as measured in the IMC which is of lower sensitivity due to higher background levels (which also can be seen for some of the markers in 3b). In addition, the segmentation algorithm is limited in precision to the resolution of the IMC as shown from the large variation in these measurements in different samples (compare the three panels in 3a). In current versions of the IMC software, the resolution has been improved by spot reduction as well as improved segmentation. The CTC shown in 3a had a particularly clean distribution of the CD45 IMC signal. In addition, the ion count plot only takes two markers (HSPD1 and CD45) into consideration. Importantly, while CD45 intensity normally varies between subsets of leukocytes from very low to strong, identification of LEVs was also based on DAPI-negativity. As demonstrated in Fig1B, the discrimination between LEVs and leukocytes were clear from the IF and brightfield imaging. The classification statement has been further clarified (p4) as the manual visual inspection of higher resolution (40X) images, to confirm/adjust the preliminary computed classification. It is already stated (p4) that any events that could not be confidently confirmed (due to e.g. small size) were excluded from the analysis. Regarding technical/biological controls, the HD-SCA assay was developed using spiked in (cell line) cells in normal blood to train the prediction models. The surrounding leukocytes provide a baseline for setting SDOM levels for what is considered a negative/positive signal for the different markers assessed for which we have referenced previous publications.    

Point 7: Previous studies have suggested, expanding exclusion criteria beyond CD45 expression increases accuracy of CTC detection. While some myeloid WBCs may bind panCK antibody aspecifically, some neutrophils and CD45- immune subsets including a subset of mature plasma cells may also express panCK specifically. This would be particularly important when analyzing/confirming EpCAM- and AR- subsets of LEVs and CTCs. Have the Authors assessed exclusion of any myeloid, endothelial, plasma cells or nucleated ertythroid progenitor markers for LEV detection?

Response 7: We agree with the reviewer and recognize that this can be an issue in discriminating liquid biopsy cells. However, in the HD-SCA analyses tumor derived circulating events in parallel and in comparison to the patients’ own leukocytes. Thus, all nucleated cells are plated and interrogated by the software at a single-cell level. The no-cell-left-behind principle in addition to the much higher resolution imagery provided for each cell on a slide facilitates the discrimination of CTCs from leukocytes. Although myeloid WBC may have a (weak) pan-CK expression, they usually differ sufficiently in size and morphology from tumor cells. Likewise, neutrophils are readily discriminated from CTCs based on size, pan-CK intensity, CD45 expression and morphology of both whole cell cytoplasm and nuclear features, which are example of measurements that are fed into the prediction algorithm. The development and validation of the HD-SCA assay is detailed in previous publications, which we referenced. For the current study, the LEV identification is additionally based on DAPI-negativity. In the majority of samples that we process, these DAPI-negative, round, pan-CK strong events are not identified. Their presence is thus highly sample dependent, correlate to CTCs and (weakly) correlate to tumor burden and bone marrow involvement, which we believe indicate that they cannot be explained by false positive detection of leukocyte phenotypes.

A sentence was added (p5) to clarify this (see previous comment). Importantly, we recognize as the reviewer suggests that there are future studies required on delineating the biology of this vesicles.

Point 8: How reliable is the detection of 1 LEV in this analysis? 

Response 8: This is an important comment and reflects an ongoing debate in the field. It is for that reason that we are only reporting the results but not suggesting a direct extension. Given that even smaller vesicle-like objects were observed in many samples but not included due to lack of confidence in their interpretation in the given magnification, there is likely room for increased sensitivity as the detection and resolution is improved. Still, threshold for LEV assessment incorporated in any potential future assays for diagnostic/predictive purposes will have to be optimized and validated.

Point 9: How many patients in the cohort represented similarly low yield (<3) of LEV and were also negative for CTC?

Response 9: This is reflected in the text (p5) “Within CTC-negative patients (23/44, 52%), at least one LEV was identified in 73% (17/23) of cases”. 7 patients were negative for both CTCs and LEVs, which has now been added to the text (p5). Another 7 CTC-negative patients had <3 LEVs.

Point 10: Was there a sufficient number of total events analyzed to ensure the statistical significance of the data in this rare-event analysis platform with such low expected abundance?

Response 10: We recognize and state (p5) that this is indeed a low abundance observation bringing with it all the challenges of statistical evaluation. Reaching statistical power is an inherent problem of CTC analysis, as CTCs indeed are rare in many cancers, which is also why we highlight the importance of the increased sensitivity of the liquid biopsy assay that the LEV identification confers. In this proof-of-concept study, we merely report on correlation to CTCs, and tendency of higher levels in patients with higher tumor load and bone marrow involvement. We clarify statistical significance was not reached for the clinical parameters assessed (p5).

Point 11: What was the distribution of various CTC and LEV counts within the patient cohort?

Response 11: Although the distribution of CTC and LEV counts is not displayed in e.g. a histogram, all values are shown in Figure 1D, which, together with the statistics provided in the text, will hopefully be sufficient to indicate the counts per patient.

Point 12: Can the Authors clarify if the data points in Figure 1 show average count between the 2 slides processed for IF or the total of 2 slides?

Response 12: We confirm and clarify in the figure legend and methods that the data points in figure 1 plot the average count of the two slides analyzed with IF. 

Point 13: Can the Authors comment on intra-patient reproducibility between the 2 slides processed under Results 3.1?

Response 13: Inter-slide variation has now been calculated and added to Results (p5). Based on patients with >5 events on average (as CV values are misleading for the many patients having low numbers with e.g. 2 on slide 1 and 0 on slide 2), the CV is 0.39 for CTCs and 0.21 for LEVs. However, it should be recognized that the original assay developed arrived at 2 slides to be a stable average, i.e. any 2 slides together are similar to the next 2 two slides from the same patient. This concordance was characterized previously and is now referenced in this paper.

Point 14: In Figure 1E, do the data points in the Tumor Load and in the BM status graphs represent the same cohort? Do the open circles represent individual data points? There seem to be fewer patients plotted on the BM graphs vs the Tumor load graphs. For example, some of the patients within ~20-30 count range seem to not be represented on the BM status graph. Can the Authors clarify? This figure would benefit from the addition of mean/median + error bars to better reflect distribution.

Response 14: Thank you for pointing out the necessary improvements to the clarity of this figure. Each circle represents an individual data points, now clarified in the legend. Data on tumor load, scored as high, intermediate, low, was available for 41/44 patients, and on BM involvement for 37/44 patients, which is stated in Table 1. One of the patients for who we lacked information on BM involvement had an average of 27 CTCs and 27 LEVs, respectively, which are the ‘missing’ data points that the reviewer refers to. A new version of the figure shows the mean + error bars.

Point 15: The counts were normalized to approximate number of nucleated cells put on slides and ROIs interrogated. Have the Authors looked at counts normalized per ml blood as the nucleated yields per ml peripheral blood could vary considerably in this patient group?

Response 15: Within the HD-SCA, the number of events is normally reported per mL of blood. With the purpose of assessing LEV correlation to CTCs on the same slides, conversion to concentrations was redundant. Concentration will however be applied in future studies, for defining clinical predictive value thresholds.

Point 16: The Authors should make the ‘CK’ abbreviation concise throughout the manuscript. CK is first mentioned in the abstract as reference to ‘cytokeratin’ in context of the IMC platform, then again used as an abbreviation on Page 3 in Materials and Methods 2.1 as reference to ‘pan-cytokeratin’.

Response 16: Pan-CK is now used throughout the text in the context of the IF assay. For the IMC, CK8/18 is used.

Point 17: On Page 3, under Methods 2.2 in the description of CTC and LEV identification, the CTCs were described as ‘pan-CK-positive’ and the LEVs were defined as ‘CK-positive’ events. The Authors should clarify if the criteria were different for the identification of CTCs and LEVs. Figure 1B and legend should also include the key. What were the specific pan-CK and CK19 clones used in the protocol included on page 3?

Response 17: The same pan-CK IF panel was used for identifying CTCs and LEVs, which has now been clarified on p3: “Using the same 3-color IF panel…”, and by changing CK to pan-CK throughout the text. The antibodies used have now been specified (p3). The legend for Figure 1B has been updated to clarify composite, arrows and size bars. 

Point 18: On Page 9, under Results 3.3 the Authors report that ‘49% were negative for EpCAM’. The Authors should clarify what population this data refers to.

Response 18: This refers to all interrogated objects, which now has been clarified: “Among all pan-CK-positive objects (CTCs, Apoptotic-like cells and LEVs),…”

Point 19: The IMC analysis presented under Results 3.4 on Page 10 shows a subset of panCK positive events from Patients 1,2,3. The Authors should clarify how this subset was selected for this analysis and how they reflect to the overall heterogeneity in Patients 1, 2 and 3. For example, the Authors report 49% of the events presented in section 3.3 do not express EpCAM, however, only 7 out of 24 objects depicted on Figure 4A seem to be EpCAM negative.

Response 19: This cluster was identified through unsupervised clustering and was included particularly because of the concurrent expression of epithelial and mesenchymal markers, which we found interesting and reflective of the heterogeneity that can be captured using single-object analysis. A clarification has now been added (p11): “…was observed in a small subset cluster of objects (Fig 4A) identified through unsupervised clustering, including events from all three patients.”. The full clustering displaying the overall heterogeneity is included as Supplementary (Figure 4).

Point 20: There is a typo on Page 5, in second paragraph ‘6x106’.

Response 20: Thank you, this is now corrected.

Reviewer 2 Report

The Authors have developed a method to identify and molecularly characterize large extracellular vesicles in parallel to circulating tumor cells at a single-cell/single-vesicle level. They show that the number of large extracellular vesicles correlates to and exceeds the number of circulating tumor cells, and that analysis of tumor-derived large extracellular vesicles may increase the sensitivity of the liquid biopsy assay.

A Minor revision is recommended.

  • The MISEV guidelines should be discussed in regard to these findings.
  • Implications of the PSMA results in prostate cancer small vesicles in Krishn et al, Matrix Biology, 2018 should be discussed.
  • May the authors try a different technique to prove expression of tetraspanins in addition to CD9, given the lack of antibodies validated for IMC?
  • The abstract should summarize the results in addition to the technical accomplishments.

Author Response

Response to Reviewer 2 Comments

The authors appreciate the reviewer’s positive comments, the meticulous review of the manuscript and the insightful and helpful comments.

Point 1: The MISEV guidelines should be discussed in regard to these findings.

Response 1: The MISEV guidelines are specifically defined to apply to separation and isolation of vesicles, which does not apply to this methodology of direct imaging.

Point 2: Implications of the PSMA results in prostate cancer small vesicles in Krishn et al, Matrix Biology, 2018 should be discussed.

Response 2: The suggested reference focuses exclusively on exosomes, and those isolated from plasma, and demonstrates the expression of the αvβ3 integrin, PSMA, CD9, CD63 in a subset. We have discussed and cited references highlighting the distinction of exosome isolation from the LEVs.

Point 3: May the authors try a different technique to prove expression of tetraspanins in addition to CD9, given the lack of antibodies validated for IMC?

Response 3: We believe that this study together with the existing body of literature on larger tumor-derived vesicles / oncosomes provides sufficient evidence that the vesicles that we identify are separate from exosomes. The tetraspanins would be included to substantiate this. IF could be used to further analyze other tetraspanins, but considering what we and others have observed, we expect the signal from these markers to be low.

Point 4: The abstract should summarize the results in addition to the technical accomplishments.

Response 4: We appreciate this perspective but want to be particular cautious so as to not run the risk of overextending the data. This study provides the initial groundwork for the LEV identification approach and the association of LEVs to CTCs was the purpose of the study. Given the small number of patients included in the IMC analysis, it serves as validation and not yet as biological mechanism.

Reviewer 3 Report

Gerdtsson and coworkers studied the relation between large extracellular vesicle and circulating tumor cells in metastatic castrate resistant prostate. The results are interesting. However, a few issues shall be addressed.

  1. The information of all the chemical (company and purity) used in this study has to be listed.
  2. Table is not well organized.
  3. For the readers, the protocol of how to extract and purify the large extracellular vesicle should be added in the main text.
  4. How about the lipid composition difference between CTC and LEVs?
  5. Page 2, Several recent reviews (doi.org/10.3390/cells9122639; doi.org/10.1016/j.actbio.2021.01.010) should be included into [2, 3].
  6. Page 3, ‘-80oC’ should be changed to ‘-80oC’.

        Page 5, ‘6x106’ should be changed to ‘6x106’.

  1. Reference: some abbreviation is not right, such as ‘P.L.o.S.’, ‘A.C.S.’.

Author Response

Response to Reviewer 3 Comments

The authors appreciate the reviewer’s positive comments, the meticulous review of the manuscript and the insightful and helpful comments.

Point 1: The information of all the chemical (company and purity) used in this study has to be listed.

Response 1: The protocol was entirely based on the HDSCA assay, which has been cited in the paper. We have added information (including dilution, manufacturer and catalogue number) of antibodies used (p3).

Point 2: Table is not well organized.

Response 2: We agree that the table was hard to read due to formatting issues. It has now been reformatted (p3). We would gladly take suggestions on how the table organization can be improved and also give the editor free hands to work on the formatting.

Point 3: For the readers, the protocol of how to extract and purify the large extracellular vesicle should be added in the main text.

Response 3: The assay is based on (fixed) plated cells (and vesicles) as described in referenced reports and does not rely on any extracting and purifying of the LEVs. We state in the text that the parallel interrogation of both CTCs and LEVs, in the context of the patients own leukocytes, is one of the strengths of the approach, which is described in the text.

Point 4: How about the lipid composition difference between CTC and LEVs?

Response 4: This is an interesting comment and something that we have not assessed. The overall similarity in protein content and brightfield inspection of the cells suggest that (apart from lacking nucleus) the composition of the intact CTCs and the smaller, non-nucleated LEVs are highly similar. We discuss the potential ways of LEV generation (p13), and suggest that they are derived from viable solid or circulating tumor cells, which also is supported by much of the cited work on e.g. oncosomes (see ref 17), showing that vesicular objects can be generated by plasma membrane blebbing from particularly aggressive tumor cells.

Point 5: Page 2, Several recent reviews (doi.org/10.3390/cells9122639; doi.org/10.1016/j.actbio.2021.01.010) should be included into [2, 3]. 

Response 5: We thank the reviewer for the suggestions. The citations have now been added (p2, p15). Note the indices of references have been shifted.

Point 6: Page 3, ‘-80oC’ should be changed to ‘-80oC’.

Response 6: Thank you. this is now corrected.

Point 7: Page 5, ‘6x106’ should be changed to ‘6x106’.

Response 7: Thank you. this is now corrected.

Point 8: Reference: some abbreviation is not right, such as ‘P.L.o.S.’, ‘A.C.S.’.

Response 8: We utilized the software generated abbreviations but will adjust based on editorial suggestions.